# Prognostic comparison of COVID-19 outpatients and inpatients treated with Remdesivr: A retrospective cohort study

Shokoofeh Zamani[1], Mohammad Alizadeh[2], Ehsan Shahrestanaki[3],
Sahar Mohammadpoor Nami[3], Mostafa Qorbani[3], Maryam Aalikhani[4], Saeed Hassani
Gelsefid[3], Nami Mohammadian Khonsari[3]*

**1** Department of Internal Medicine, School of Medicine, Imam Ali Hospital, Alborz University of Medical
Sciences, Karaj, Iran, **2** Department of Infectious Diseases, School of Medicine, Imam Ali Hospital, Alborz
University of Medical Sciences, Karaj, Iran, **3** Non-Communicable Diseases Research Center, Alborz
University of Medical Sciences, Karaj, Iran, **4** Student Research Committee, Alborz University of Medical
Sciences, Karaj, Iran

* nami.m.kh@gmail.com

## Abstract

### Introduction

Since the late COVID-19, many countries have faced various surges and peaks within the number of infected. Iran was one of the countries that faced many surges and peaks within these years and faced many inadequacies and shortages of resources and hospital beds. Hence the healthcare system started using in-hospital medication such as Remdesivir in outpatients to reduce the load of patients admitted to the hospital. This study aimed to evaluate and compare the reported signs, symptoms, and outcomes of COVID-infected hospitalized and out-patients receiving Remdesivir.

### Methods

In this retrospective cohort study, 214 patients (121 outpatient and 93 hospitalized) with moderate levels of Covid infection between October 2021 and February 2022 were studied. Both groups were treated with 200mg of Remdesivir, followed by 100 mg daily intravenous injections for five days; signs and symptoms, such as pain, shortness of breath, cough, fever and etc., of patients at the initiation and the end of treatment were recorded. Moreover, the patients' blood oxygen saturation was assessed two to three times a day, and the mean of the recorded measures was considered as the daily oxygen saturation. The outpatient group had to visit the hospital daily for treatment and assessment. At the treatment's end, mortality rates, disease signs, and symptoms alleviations were compared between the groups.

### Results

The outpatient and hospitalized group's mean age was 40.30 ± 12.25 and 37.70 ± 12.00 years, and 51.2% and 55.9% were males, respectively. There was no statistical difference between baseline and clinical characteristics in the outpatients and hospitalized groups.

**Data Availability Statement:** All relevant data are available from the Zenodo repository (https://doi. org/10.5281/zenodo.7262680).

**Funding:** Alborz University of Medical Sciences funded this study the fund was given to Dr SH.Z The funders had no role in study design, data collection and analysis, decision to publish, or preparation of the manuscript.

**Competing interests:** The authors declare that they have no conflict of interest.

After adjusting for oxygen saturation at baseline and gender in the multivariable Cox regression analysis, the risk of death did not statistically differ between the hospitalized and outpatient group (hazard ratio: 0.99, 95% confidence interval: 0.39–2.50)) at the end of the study.

## Conclusion

Based on the results of this study, the outcome, signs, and symptoms of inpatient and outpatient Remdesivir treatment groups did not differ significantly. Hence in COVID-19 surges where we have limitations in admitting patients, outpatient Remdesivir treatment for those without any underlying diseases can be a proper management method.

## Introduction

Since the late COVID-19 pandemic, many countries have endured severe casualties [1]. Despite protective measures, treatments, and even vaccinations, surges in the number of infected were seen [2–4]. Within these surges, the number of infected increased drastically, and countries faced insufficient hospital beds, personnel, and resources. This inadequacy is of great importance since it can result in massive mortalities and extra expenses for the healthcare system and the patients; moreover, this shortage of services and supplies affects hospital referees other than the COVID infected as well [5]. With the outbreaks of various subtypes of the COVID-19 virus, countries worldwide experienced several peaks and surges within the infected [2, 3]. Iran was one of the countries that faced many surges and peaks within these years, and many difficulties arose from the vast number of the infected [6]. Hence in these desperate times, desperate measures had to be taken. One of these measures was using in-hospital medication such as Remdesivir in outpatients to reduce the load of patients admitted to the hospital. Remdesivir is a broad-spectrum antiviral agent that showed some promise in treating COVID patients. This drug is administered via intravenous injection and was used solely as an in-patient drug [7].

Nonetheless, under the circumstances, Iran's healthcare system had no choice but to design an outpatient protocol for Remdesivir use, and with this scheme, the number of hospitalizations due to COVID-19 dropped dramatically. However, this decision was solely based on the shortage of personnel, supplies, and hospital beds, and no study was done on this matter to compare and evaluate the therapeutic effects and the outcome of the use of this drug among admitted and out-patients. Iran was among the very first countries that approved the use of this drug as an outpatient drug. Thus this study aimed to evaluate and compare the reported signs and symptoms and the outcome of COVID-infected hospitalized and out-patients receiving Remdesivir in a retrospective cohort study.

## Methods

This retrospective cohort study assessed all the COVID patients referred to the Imam Ali hospital of Alborz university of medical sciences, Karaj, Iran, from first October 2021 until the end of February 2022.

### Sampling

Due to the retrospective nature of this study and the significance of possible confounders, we adapted explicit inclusion and exclusion criteria to homogenize the hospitalized and

outpatients for a more precise comparison. The included patients had to be between the ages of 20 to 60, with an oxygen saturation between 88 to 93% and within the first seven days of the initiation of symptoms. Moreover, having a positive polymerase chain reaction (PCR) test alongside a Computed tomography (CT) scan of the lungs was necessary. Only patients with bilateral round morphology ground-glass opacities affecting three to four pulmonary lobes without any signs of pleural effusion, cavitation, fibrosis, and bronchiectasis were included. The participants had to be nonsmokers without any underlying conditions, diabetes or hypertension, and obesity. The patients who took medication of any kind, seven days prior to their referral were excluded from the study. All patients had to address and follow the exact same treatment protocol. Only those vaccinated with the first dose of the Sinopharm COVID vaccine were included in the study. The grouping of the participants was solely based on the availability of hospital beds. If hospital beds were not available for our referral patients, they had the choice to be treated as outpatients or go to another hospital. We ensured that outpatients included in this study had proper home care and a medical oxygen cylinder at home.

## Sample size

The sample size was determined according to previous studies [8, 9]. By considering alpha and beta errors of 0.05 and 0.2, respectively, and a mortality risk in the hospitalized and outpatients 11% [8] and 1% [9] (risk difference of 10%) the sample size was determined 178 subjects using G-power program [10] (89 participanrs in each group).

## Data collection

All the patients underwent a thorough medical assessment, and all patients who fell within our inclusion criteria were included in this study. The age and sex of the patients were recorded. Their initial symptoms, including coughs, presence of any pain, shortness of breath, and vital signs at the time of referral, were recorded. In the admitted patients, the oxygen saturation was assessed with a pulse oximeter every six hours, four times a day, and the mean of the recorded values was considered the daily oxygen saturation. Similarly, for the out-patients, the oxygen was measured with a pulse oximeter when they visited the hospital for their Remdesivir treatment and six hours after Remdesivir administration. The mean of the recorded values was considered as their oxygen saturation. We assessed our patients' shortness of breath by categorizing it into four subgroups. No shortness of breath during ordinary daily activity, mild shortness of breath during routine daily activity, shortness of breath during routine daily activity; however, being at ease while resting (this was considered as moderate shortness of breath), and shortness of breath at all times (this was considered as severe shortness of breath). To assess pain, we asked our participants to rate their current pain (including headache, backache, muscle ache, etc.) from 0, meaning the absence of any pain, to 10, meaning the worst pain they ever endured. Pain evaluation was done three times (once at the beginning of treatment, the second at the end of Remdesivir treatment, and the third time when the patient fully recovered). Body temperature was assessed using an analogue medical thermometer during medical examinations. An oral recorded temperature above 37.5 degrees Celsius (C) was considered as the presence of fever. For analytic purposes, we categorized fevers into four subgroups. The absence of fever, mild (37.5–38.5) C, moderate (38.6–39.5) C, and severe fever (39.6 C and above). Moreover, we asked our patients about the frequency and severity of their coughs at the end of treatment and how they had changed during this period. The patients within the two groups were followed until recovery, or if they had passed away, the need for hospitalization (in cases of outpatients), or were sent to the intensive care unit (ICU) despite medical treatment.

Recovery was defined as having a stable oxygen saturation of > 92% within the past twelve hours and having no severe coughs, shortness of breath, and absence of fever.

Outpatients hospitalized due to exacerbation of their condition were still considered as part of the outpatient group.

Remdesivir treatment was continued until the patient had partially recovered (stable oxygen saturation of > 92% within the past 24 hours without an oxygen mask/nasal cannula and having no severe coughs, shortness of breath, and fever.), passed away due to COVID, or completed the five-day duration of Remdesivir treatment.

## Treatment protocol

The primary treatment protocol was as follows during Remdesivir therapy and was carried out for all patients. 1) oxygen with a mask or nasal cannula if needed 2) intravenous dexamethasone 8mg daily for admitted patients and oral prednisolone 40mg daily for outpatients 3) enoxaparin, subcutaneous injection, once daily. 4) at the first day of treatment, 200mg of Remdesivir was given to the patients, followed by 100 mg daily intravenous injections for five days. The 100mg vials of Remdesivir were diluted with 200ml of normal saline and administered intravenously for 30 minutes.

## End points

The primary endpoint of this study was COVID-related death or disease-related symptoms exacerbation needing hospitalization in the outpatient group and COVID-related death or ICU admission in the hospitalized group during the Remdesivir treatment course (five days) and then during follow-up till 28 days post-infection. (based on similar studies [9]).

The secondary endpoint was the comparison of disease alleviation duration and oxygen saturation changes (during Remdesivir treatment) among the hospitalized and outpatients.

Our third endpoint was comparing the full recovery time between the two groups.

Our safety outcome was severe or potentially life-threatening Remdesivir treatment adverse effects leading to treatment discontinuation of Remdesivir treatment. Severe adverse effects were defined based on the division of AIDS (DAIDS) table for Grading the Severity of Adult and Pediatric Adverse Events version 2.1 [11].

## Ethical consideration

We explained the study to those who met the inclusion criteria or their first-degree relatives (if they had passed away) in full and written consent was obtained regarding the use of their data for research and publication purposes. Furthermore, the patients were assured that their personal information would not be shared with anyone throughout this study. Moreover, the Ethics Committee of Alborz University of Medical Sciences approved this study (No.IR. ABZUMS.REC.1400.306). All methods of the study were carried out by relevant guidelines and regulations.

## Statistical analysis

By the Kolmogorov-Smirnov test, we evaluated the normal distribution of continuous variables. (oxygen saturation, pain, and age). Categorical variables were expressed as frequency and percentage (n (%)) and continuous variables as mean and standard deviation (SD).

Repeated measures analyses were used to compare the variables at the initiation and end of the treatment. Univariable and multivariable (adjusted for age, gender, oxygen saturation and etc.) cox regression analysis was used to compare the mortality risk between the two treatment

groups. We reported the cox regression analysis as hazard ratio (HR) and 95% confidence interval (CI). A two-tailed P-value less than 0.05 was considered statistically significant. SPSS (Statistical Package for the Social Sciences software, version 25) was used for data analyses.

## Results

In total, 214 patients diagnosed with covid-19 were included in the study, consisting of 121 (56.50%) outpatients and 93 (43.45%) hospitalized. Table 1 summarises the basic and clinical characteristics of the study population. The outpatients and hospitalized group's mean age was 40.30 ± 12.25 and 37.70 ± 12.00 years, and 51.2% and 55.9% were males, respectively. There was no statistical difference between baseline and clinical characteristics in the outpatients and hospitalized groups. (p-value>0.05), except for shortness of breath (the number of moderate and severe cases of shortness of breath in the hospitalized group was greater than in the outpatient group (P-value: 0.005)).

At the end of the Remdesivir treatment (day 5), the remnant pain and cough in the outpatient group were statistically more significant than in the hospitalized group. However, there was no statistical difference between the severity of shortness of breath and fever in outpatient and hospitalized groups. 9.92% and 8.60% of deaths occurred in outpatient and hospitalized groups, respectively, which was not statistically significant.

In the univariable and multivariable cox regression analysis, the risk of mortality was not statistically different in the hospitalized patients compared with the outpatient group ((non-adjusted HR: 0.81, (95%CI: 0.33, 2.00)) adjusted HR: 0.99, (95% CI: 0.39–2.50)) (Table 2).

Based on repeated measures analysis, the mean oxygen saturation was not statistically different between the outpatients and hospitalized groups during treatment time (Table 3). However, the mean oxygen saturation was statistically different between the living and the dead during the treatment period (Table 4).

**Table 1. Basic characteristics of hospitalized covid-19 patients categorized into outpatients and hospitalized.**

| Variables | | Outpatient | Hospitalized | P-value |
|---|---|---|---|---|
| | | n = 121 | n = 93 | |
| Age (Mean ± SD) | | 40.30 ± 12.25 | 37.70 ± 12.00 | 0.119 |
| Pain (Mean ± SD) | | 5.45 ± 1.78 | 5.36 ± 2.35 | 0.766 |
| Oxygen saturation (measured by pulse oximetry) | | 90.90 ± 1.40 | 90.55 ± 1.50 | 0.069 |
| Gender n (%) | Male | 62 (51.20) | 52 (55.90) | 0.497 |
| | Female | 59 (48.80) | 41 (44.10) | |
| Cough | No | 60 (49.60) | 37 (39.80) | 0.153 |
| | Yes | 61 (52.10) | 56 (47.90) | |
| Shortness of breath n (%) | No | 26 (21.50) | 11 (11.80) | 0.005* |
| | Mild | 53 (43.80) | 32 (34.40) | |
| | Moderate | 33 (27.30) | 29 (31.20) | |
| | Sever | 9 (7.40) | 21 (22.60) | |
| Fever | No | 14 (11.60) | 7 (7.50) | 0.608 |
| | Mild | 37 (30.60) | 29 (31.20) | |
| | Moderate | 53 (43.80) | 39 (41.9) | |
| | Severe | 17 (14.0) | 18 (19.40) | |

n: number, %: percentage, SD: standard deviation.

* statistically significant (P-value < 0.05).

**Table 2. The risk of mortality according to type of treatment.**

| Variables | Outpatient | Hospitalized | Crude hazard ratio | Adjusted hazard ratio Model I | Adjusted hazard ratio Model II | Adjusted hazard ratio Model III |
|---|---|---|---|---|---|---|
| | n = 121 | n = 93 | (95% CI) | (95% CI) | (95% CI) | (95% CI) |
| Dead | 12 (9.90) | 8 (8.60) | 0.81 (0.33, 2.00) | 0.99 (0.39, 2.50) | 0.83 (0.32, 2.12) | 0.93 (0.34, 2.53) |

Model I: The multivariable Cox regression was adjusted for age, gender, and oxygen saturation at baseline.

Model II: additionally adjusted for baseline shortness of breath.

Model III: additionally adjusted for baseline pain, cough, and fever.

n: number, CI: confidence interval.

**Table 3. Comparison of oxygen saturation means and standard deviations (Mean ± SD) between hospitalized and outpatients during Remdesivir treatment.**

| Variable | Outpatient | Hospitalized | P-value[1] | Effect size |
|---|---|---|---|---|
| Day 1 | 90.90 ± 1.40 | 90.55 ± 1.50 | 0.124 | 0.010 |
| Day 2 | 90.80 ± 1.88 | 89.50 ± 3.25 | | |
| Day 3 | 90.62 ± 2.32 | 89.72 ± 2.74 | | |
| Day 4 | 90.90 ± 2.91 | 89.75 ± 3.70 | | |
| Day 5 | 91.45 ± 3.78 | 91.00 ± 3.87 | | |

The P-value and effect size were acquired using repeated measures analysis after applying the Bonferroni correction

SD: standard deviation.

**Table 4. Comparison of the means and standard deviations of oxygen saturation (Mean ± SD) between patients who survived and passed away due to Covid during Remdesivir treatment.**

| Variable | Alive | dead | P-value[1] | Effect size |
|---|---|---|---|---|
| Day 1 | 90.73 ± 1.40 | 91.05 ± 1.68 | <0.001* | 0.351 |
| Day 2 | 90.35 ± 2.59 | 88.90 ± 2.76 | | |
| Day 3 | 90.47 ± 2.30 | 87.73 ± 3.47 | | |
| Day 4 | 91.00 ± 2.30 | 84.31 ± 5.48 | | |
| Day 5 | 92.12 ± 2.14 | 82.21 ± 5.33 | | |

* statistically significant (P-value < 0.05).

The P-value and effect size were acquired using repeated measures analysis after applying the Bonferroni correction

SD: standard deviation.

Moreover, the duration of full recovery time (median ± interquartile range) among the hospitalized and outpatients was 5 ± 2 and 6 ± 1 days, respectively; which was statistically significant (P-value: 0.01).

Regarding our safety outcomes, none of the patients experienced severe or potentially life-threatening adverse events.

## Discussion

Based on our results, outpatients and inpatients did not significantly differ in mortality rates, disease signs, and symptom alleviation. Hence based on our results, outpatient Remdesivir treatment can be as effective as inpatient Remdesivir treatment in patients without any underlying diseases. Similar to our findings, another study found that the outcome and mortality rates of inpatients and outpatients treated with Remdesivir did not differ significantly [12].

Although there have been controversies regarding the Covid-19 treatment protocol since the pandemic's initiation, most studies concur that early treatment significantly reduces severity and mortality rates [9, 12]. Moreover, even though the effectiveness of Remdesivir treatment (in hospital) is under question, many studies have shown that early Remdesivir treatment reduces the adverse outcomes of COVID, and some have suggested that early outpatient treatment with Remdesivir has reduced the severity and mortality among Covid patients [9].

We found that full recovery time among the hospitalized patients was shorter than the outpatients. Although a recovery duration difference of one day may not seem clinically significant, it was statistically significant; however, this significance cannot solely be attributed to Remdesivir treatment, as hospital care could also play a role in the patients' recovery time. In this regard, although it is out of the question that many patients need hospitalization due to COVID, we can initiate early outpatient Remdesivir treatment to cover a vast infected population, thus reducing hospitalizations and making room for those who need to be admitted or those with undelaying diseases.

In our study, the mortality rate of hospitalized patients was similar to the mortality rate reported by others [8]; however, the mortality rate of our outpatients was higher compared to similar studies [9]. The higher mortality rate in our outpatients could be attributed to the fact that our outpatients were hospitalization candidates with a more severe form of infection compared to other studies that evaluated patients who were not hospitalization candidates. Hence due to higher mortality rates in the outpatients, the power of our study may have been insufficient to detect the mortality difference between the two study arms.

It should be kept in mind that we did not evaluate participants with underlying diseases. As reported in the literature, underlying diseases tend to increase the severity and mortality of COVID infection [13–15], and we do not recommend treating COVID patients with underlying diseases as outpatients. Nonetheless, the nature, extent, and severity of the underlying diseases are major factors affecting COVID severity and mortality [16]. Thus, to safely determine the efficacy of outpatient COVID treatment, studies with large samples alongside medical teams available 24 hours a day (to prevent or treat any unwanted adverse effects in the outpatients) are needed. Moreover, there are ethical aspects to be addressed as well; such studies can take place in countries with limited hospital beds and dire situations.

Moreover, it must be noted that precise blood oxygen monitoring is recommended in both hospitalized and outpatients; since, as can be seen in Table 4, in both groups, an oxygen decline, especially a notable oxygen decline between days 3 and 4 of Remdesivir treatment, was associated with a significant mortality rate. Thus we think that regardless of the patient's condition, in cases of a decline of blood oxygen saturation, especially within days 3 and 4 in all patients, ICU admission and preventive measures must be taken into account.

Another crucial point that should be kept in mind is the side effects and adverse outcomes of Remdesivire treatment. As some studies indicate, up to 60% of the patients can show some adverse effects [17]. Although most of these adverse effects are insignificant, serious adverse events must be managed [17]. These adverse effects are manageable in admitted patients, However, managing these symptoms in outpatients can be challenging, and the patients need proper education regarding these symptoms.

## Limitations and strength

Although our sample size was greater than similar studies, more studies with a greater sample size are needed to compare the differences between inpatient and outpatient Remdesivir treatment. However, regardless of our precise sampling method, the role of potential confounders cannot be ignored; hence, randomized control trials (RCTs) are needed to determine the exact

differences between hospitalized and outpatients treated with Remdesivir, free of residual confounding.

It should be kept in mind that our study was a single-center study, and our findings can only be attributed to moderate infection levels, without any underlying diseases and in dire situations with vast numbers of infected and limited resources with the possibility of residual confounding.

To impute these findings on different situations and the general population, multicenter studies with a larger group of the infected are recommended.

## Conclusion

Based on the results of this study, the outcome, signs, and symptoms of inpatient and outpatient Remdesivir treatment groups did not differ significantly. Hence in COVID-19 surges where we have limitations in admitting patients, outpatient Remdesivir treatment, if necessary, can be a proper management method for patients without any underlying diseases.

## Author Contributions

**Conceptualization:** Shokoofeh Zamani, Mohammad Alizadeh.

**Data curation:** Nami Mohammadian Khonsari.

**Formal analysis:** Ehsan Shahrestanaki, Mostafa Qorbani, Nami Mohammadian Khonsari.

**Funding acquisition:** Shokoofeh Zamani.

**Investigation:** Mohammad Alizadeh, Sahar Mohammadpoor Nami, Maryam Aalikhani, Saeed Hassani Gelsefid, Nami Mohammadian Khonsari.

**Methodology:** Ehsan Shahrestanaki, Nami Mohammadian Khonsari.

**Project administration:** Shokoofeh Zamani, Mohammad Alizadeh, Nami Mohammadian Khonsari.

**Resources:** Nami Mohammadian Khonsari.

**Software:** Ehsan Shahrestanaki.

**Supervision:** Mohammad Alizadeh.

**Validation:** Sahar Mohammadpoor Nami, Mostafa Qorbani.

**Writing – original draft:** Nami Mohammadian Khonsari.

**Writing – review & editing:** Nami Mohammadian Khonsari.

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
