## [Decision Letter · Decision Letter 0]

30 May 2022

PONE-D-22-12580Prognostic comparison of COVID-19 outpatients and inpatients treated with Remdesivr: a retrospective cohort study.PLOS ONE

Dear Dr. Mohammadian Khonsari,

Thank you for submitting your manuscript to PLOS ONE. After careful consideration, we feel that it has merit but does not fully meet PLOS ONE’s publication criteria as it currently stands. Therefore, we invite you to submit a revised version of the manuscript that addresses the points raised during the review process.

We look forward to receiving your revised manuscript.

Kind regards,

Tai-Heng Chen, M.D.

Academic Editor

PLOS ONE

4. Please include a copy of Table 3 which you refer to in your text on page 7.

Reviewers' comments:

Reviewer's Responses to Questions

**Comments to the Author**

1. Is the manuscript technically sound, and do the data support the conclusions?

Reviewer #1: Partly

Reviewer #2: Yes

2. Has the statistical analysis been performed appropriately and rigorously? 

Reviewer #1: No

Reviewer #2: Yes

3. Have the authors made all data underlying the findings in their manuscript fully available?

Reviewer #1: Yes

Reviewer #2: Yes

4. Is the manuscript presented in an intelligible fashion and written in standard English?

Reviewer #1: Yes

Reviewer #2: Yes

5. Review Comments to the Author

Reviewer #1: I read with interest the study by Shokoofeh Zamani et al. entitled “Prognostic comparison of COVID-19 outpatients and inpatients treated with Remdesivr: a retrospective cohort study”. The study compared outcomes of the two groups treated with Remdesivr. I am sorry that the study can’t be accepted. Below you may find my comments:

1. Authors should acknowledge one of the study’s limitations is the limited extrapolation of conclusions caused by strict inclusion and exclusion criteria. Moreover, most of the death events occurred in people with more comorbidities, and this study excluded people with underlying diseases. These should be at least addressed within a paragraph in the discussion.

2. The endpoints of the study should to be added, such as viral load. Besides, safety outcomes also should be reported.

3. Cox regression method is more suitable for comparing mortality differences.

4. “2.5 Statistical analysis”: Mind that the term multivariate logistic regression refers to multiple logistic regression or multivariable logistic regression; please amend. Of note, the reliability of the conclusion is doubtful because of few factors for model adjustment.

Reviewer #2: The present study aimed to evaluate and compare the reported signs and symptoms, and the outcome of COVID-infected hospitalised and out-patients receiving Remdesivir. Although the research question is interesting, I have some comments:

Major remarks

1) Randomized clinical trials have shown an positive effect of Remdesivir on shortening the time to recovery in adults who were hospitalized with Covid-19, without significant impact on mortality. In this sense, it would be interesting if the authors could compare time to recovery between the two cohort of patients (hospitalized vs. non-hospitalized patients).

2) Since the main result of this study was neutral, the analysis of type II error is crucial. Please explain how sample size was determined. A power analysis showing the actual power of the present study would be informative for readers.

3) It seems that the present study is underpowered for a categorical outcome such as mortality. I might suggest to reinforce throughout discussion and conclusions the fact that the present study result may be due to the lack of power.

4) I might suggest to include shortness of breath at baseline as an covariate in the multivariate logistic model assessing the comparison of the two cohorts on the main outcome.

5) Why the authors did no use Cox regression to assess mortality?

Minor remarks

1) Please include footnotes in tables in order to describe the statistic used (e.g., data are n(%)) and abbreviations;

2) Please provide information in table footnotes to help readers to interpret the outcomes assessed.

3) Please explain how effect size was calculated.

4) Please revise the term oxygen in table 1 (maybe replace by oxygen saturation measured by pulse oximetry)

6. PLOS authors have the option to publish the peer review history of their article (what does this mean?). If published, this will include your full peer review and any attached files.

Reviewer #1: No

Reviewer #2: **Yes: **Regis Goulart Rosa

---

## [Author Response · Author response to Decision Letter 0]

1 Jul 2022

First I would like to thank the reviewers and editor. Their comments improved the manuscript significantly, and for that, I’m grateful 

The entire manuscript has been re-read. Some changes within sentences were made and some grammatical corrections were made to improve readability. 

Moreover, all the comments were addressed and changes are highlighted 

Editorial comments:

1. The referencing style has been edited and corrected (square brackets)

2. The ethics section has been moved to methods

3. The tables have been edited and corrected. 

4. regarding data availability, accession numbers and/or DOIs will be made available after acceptance (the data file will be made public if accepted, prior to publication)

Reviewer #1

1. Authors should acknowledge one of the study’s limitations is the limited extrapolation of conclusions caused by strict inclusion and exclusion criteria. Moreover, most of the death events occurred in people with more comorbidities, and this study excluded people with underlying diseases. These should be at least addressed within a paragraph in the discussion.

thank you for the point

The term strict may have been miss interrupted; hence it was replaced with “ explicit” as we meant to say we used precise inclusion and exclusion criteria. 

We agree with the comment, we had some limitations however we had acknowledged our limitations throughout the discussion.

 Nonetheless, we have further elaborated our limitations based on the comment 

2. The endpoints of the study should to be added, such as viral load. Besides, safety outcomes also should be reported.

thank you for the comment. the requested points were added under “End points”

3. Cox regression method is more suitable for comparing mortality differences.

Thank you for the very important point, Cox regression was done and the results of logistic regression were substituted with Cox regression

4. “2.5 Statistical analysis”: Mind that the term multivariate logistic regression refers to multiple logistic regression or multivariable logistic regression; please amend. Of note, the reliability of the conclusion is doubtful because of few factors for model adjustment.

thank you for this very important point, corrections were made throughout the entire manuscript 

moreover based upon your request, we made some changes (Cox regression model III: adjusted for : sex, age, baseline pain, shortness of breath, cough, fever and oxygen saturation.

Reviewer #2:

Major remarks

1) Randomized clinical trials have shown an positive effect of Remdesivir on shortening the time to recovery in adults who were hospitalized with Covid-19, without significant impact on mortality. In this sense, it would be interesting if the authors could compare time to recovery between the two cohort of patients (hospitalized vs. non-hospitalized patients).

Thank you for the comment, the requested data were added

2) Since the main result of this study was neutral, the analysis of type II error is crucial. Please explain how sample size was determined. A power analysis showing the actual power of the present study would be informative for readers.

Thank you for the point, sample size calculation was added 

3) It seems that the present study is underpowered for a categorical outcome such as mortality. I might suggest to reinforce throughout discussion and conclusions the fact that the present study result may be due to the lack of power.

Although the power of this study was set to 80%, (added in the sampleing and sample size) the discussion has been improved. 

4) I might suggest to include shortness of breath at baseline as an covariate in the multivariate logistic model assessing the comparison of the two cohorts on the main outcome.

Thank you for the comment, although shortness of breath seems important as a covariate and we wanted to include it in the multivariate analysis, we thought it is rather a subjective variable with no definitive measurement, and patients definition and sensation on the subject may differ, hence the data on shortness of breath completely relied on patient reports, thus we thought it would be better to use oxygen saturation ( a definitive and measurable variable, more precise and could affect shortness of breath as well) as a covariate for a proper analysis a variable. 

However due to your kind request we added the results of analysis with shortness of breath as a covariate as well, (Cox regression model II)

5) Why the authors did no use Cox regression to assess mortality?

Thank you for the very important point, Cox regression was done and the results of logistic regression were substituted with Cox regression

 Minor remarks

1) Please include footnotes in tables in order to describe the statistic used (e.g., data are n(%)) and abbreviations;

Thank you for the point 

done 

2) Please provide information in table footnotes to help readers to interpret the outcomes assessed.

Thank you for the point 

done 

3) Please explain how effect size was calculated.

Thank you for the point. It was added in the table footnote

4) Please revise the term oxygen in table 1 (maybe replace by oxygen saturation measured by pulse oximetry)

Thank you for the point 

done

---

## [Decision Letter · Decision Letter 1]

1 Sep 2022

PONE-D-22-12580R1Prognostic comparison of COVID-19 outpatients and inpatients treated with Remdesivr: a retrospective cohort study.PLOS ONE

Dear Dr. Mohammadian Khonsari,

Thank you for submitting your manuscript to PLOS ONE. After careful consideration, we feel that it has merit but does not fully meet PLOS ONE’s publication criteria as it currently stands. Therefore, we invite you to submit a revised version of the manuscript that addresses the points raised during the review process. Please submit your revised manuscript by Oct 16 2022 11:59PM. If you will need more time than this to complete your revisions, please reply to this message or contact the journal office at plosone@plos.org. Please include the following items when submitting your revised manuscript:A rebuttal letter that responds to each point raised by the academic editor and reviewer(s). You should upload this letter as a separate file labeled 'Response to Reviewers'.A marked-up copy of your manuscript that highlights changes made to the original version. You should upload this as a separate file labeled 'Revised Manuscript with Track Changes'.An unmarked version of your revised paper without tracked changes. You should upload this as a separate file labeled 'Manuscript'.If applicable, we recommend that you deposit your laboratory protocols in protocols.io to enhance the reproducibility of your results. Protocols.io assigns your protocol its own identifier (DOI) so that it can be cited independently in the future. For instructions see: https://journals.plos.org/plosone/s/submission-guidelines#loc-laboratory-protocols. Additionally, PLOS ONE offers an option for publishing peer-reviewed Lab Protocol articles, which describe protocols hosted on protocols.io. Read more information on sharing protocols at https://plos.org/protocols?utm_medium=editorial-email&utm_source=authorletters&utm_campaign=protocols.

We look forward to receiving your revised manuscript.

Kind regards,

Tai-Heng Chen, M.D.

Academic Editor

PLOS ONE

Journal Requirements:

Reviewers' comments:

Reviewer's Responses to Questions

**Comments to the Author**

1. If the authors have adequately addressed your comments raised in a previous round of review and you feel that this manuscript is now acceptable for publication, you may indicate that here to bypass the “Comments to the Author” section, enter your conflict of interest statement in the “Confidential to Editor” section, and submit your "Accept" recommendation.

Reviewer #1: All comments have been addressed

Reviewer #2: All comments have been addressed

2. Is the manuscript technically sound, and do the data support the conclusions?

Reviewer #1: Yes

Reviewer #2: Yes

3. Has the statistical analysis been performed appropriately and rigorously? 

Reviewer #1: Yes

Reviewer #2: Yes

4. Have the authors made all data underlying the findings in their manuscript fully available?

Reviewer #1: Yes

Reviewer #2: Yes

5. Is the manuscript presented in an intelligible fashion and written in standard English?

Reviewer #1: Yes

Reviewer #2: Yes

6. Review Comments to the Author

Reviewer #1: (No Response)

Reviewer #2: The revised version of the manuscript improved a lot. However, some points need to be addressed:

1- There are some inconsistencies regarding writing. In this sense, i might suggest to edit the manuscript for grammar and scientific style.

2- I still have some doubts regarding the sample size calculation. The sample size seems to be very low to detect a difference of 10% in mortality between the two cohorts. Please describe the expected number (%) of events (with references) in the control arm in the sample size description. Additionally, include in the discussion the fact that the study was underpowered to detect differences < 10% between the two study arms.

7. PLOS authors have the option to publish the peer review history of their article (what does this mean?). If published, this will include your full peer review and any attached files.

Reviewer #1: No

Reviewer #2: No

---

## [Author Response · Author response to Decision Letter 1]

13 Sep 2022

thank you for your thorough evaluation and comments

Dear reviewer regarding the comment:

C1: still have some doubts regarding the sample size calculation. The sample size seems to be very low to detect a difference of 10% in mortality between the two cohorts. Please describe the expected number (%) of events (with references) in the control arm in the sample size description. Additionally, include in the discussion the fact that the study was underpowered to detect differences < 10% between the two study arms.

Answer: we added the references and the issues were addressed

---

## [Editor Report · Decision Letter 2]

27 Oct 2022

Prognostic comparison of COVID-19 outpatients and inpatients treated with Remdesivr: a retrospective cohort study.

PONE-D-22-12580R2

Dear Dr. Mohammadian Khonsari,

We’re pleased to inform you that your manuscript has been judged scientifically suitable for publication and will be formally accepted for publication once it meets all outstanding technical requirements.

Kind regards,

Tai-Heng Chen, M.D.

Academic Editor

PLOS ONE
---

## [Editor Report · Acceptance letter]

2 Nov 2022

PONE-D-22-12580R2 

Prognostic comparison of COVID-19 outpatients and inpatients treated with Remdesivr: a retrospective cohort study. 

Dear Dr. Mohammadian Khonsari:

I'm pleased to inform you that your manuscript has been deemed suitable for publication in PLOS ONE. Congratulations! Your manuscript is now with our production department. 

Kind regards, 

on behalf of

Dr. Tai-Heng Chen 

Academic Editor

PLOS ONE